# Does Exercise-Induced Hypoalgesia Depend on Exercise Duration?

**DOI:** 10.3390/biology12020222

**Published:** 2023-01-30

**Authors:** Fabian Tomschi, Luisa Kieckbusch, Julius Zachow, Thomas Hilberg

**Affiliations:** Department of Sports Medicine, University of Wuppertal, Moritzstraße 14, 42117 Wuppertal, Germany

**Keywords:** pain, physiology, pressure pain threshold, bicycle ergometer, pain inhibition, adults

## Abstract

**Simple Summary:**

Physical or sports activities are believed to reduce pain sensitivity following activity sessions. Several studies have explored the effect of different exercise intensities of the activity sessions on pain sensitivity and most studies reported that pain sensitivity is reduced after the sessions. Above all, this phenomenon is observed when a rather high intensity was used. However, almost no studies have explored the effect of longer duration exercise sessions on this phenomenon. Therefore, this study aimed to explore pain sensitivity following three differently long, i.e., 30, 45, and 60 min, exercise sessions, which were conducted using the same sub-maximal intensity. The results show that none of these sessions led to differences in pain sensitivity. The results were not as expected but might lead to the assumption that the intensity might be more important for inducing a reduced pain sensitivity than the duration of the session.

**Abstract:**

Acute physical activity is assumed to lead to exercise-induced hypoalgesia (EIH). Yet, little research has been conducted dealing with the influence of exercise duration on EIH. The aim of this study was to investigate the effects of three different exercise durations using the same intensity compared to a control session on EIH. A total of 36 participants conducted three different exercise sessions on a bicycle ergometer for 30, 45, and 60 min, respectively, in addition to a passive control session. The intensity was set to 75% of the individual’s VO_2_max. Pre and post exercise, pain sensitivity was measured employing pressure pain thresholds (PPT) at the elbow, knee, and ankle joints, as well as the sternum and forehead. In addition, the conditioned pain modulation (CPM) response was conducted pre and post exercise. The results reveal that the exercises neither led to any changes in PPT measured at any landmark nor induced any CPM response effects. These results do not confirm the hypoalgesic effects usually observed after exercise. The reasons explaining these results remain rather elusive but might be explained by the low intensities chosen leading to a milder release of pain inhibiting substances, the landmarks employed for PPT measurements, or potential non-responsiveness of participants.

## 1. Introduction

Physical activity is well known to influence pain and pain perception in healthy as well as chronic pain populations [1]. An acute reduction in pain sensitivity following physical activity is known as exercise-induced hypoalgesia (EIH). This phenomenon is observed using a variety of experimental pain testing modalities [2] and EIH seems to occur in the healthy population more constantly compared to chronic pain populations [2], such as in patients with chronic musculoskeletal pain [3]. Research suggests that different exercise regimens potentially lead to EIH following exercise. Aerobic exercises, dynamic resistance exercises, as well as isometric exercises were shown to increase pain thresholds and, thus, reduce pain sensitivity up to 20% compared to a resting control condition [4]. With regard to EIH induced by aerobic exercise it is suggested that a certain intensity and a certain duration is needed to induce EIH. It is generally believed that the intensity used needs to exceed 75% of the individual’s VO_2_max and that the duration of the exercise must last for at least 10 min [5]. The exact mechanisms underlying EIH remain still fairly elusive, but some mechanisms are often discussed. The most widely considered mechanism is that exercise results in the activation of the endogenous opioid system, which is known to alter pain perception during and following exercise. Endogenous opioid-related substances such as beta-endorphins are expressed at central sites in the nervous system but are also locally expressed at exercising body parts. This may reduce pain sensitivity by attenuating the nociceptive signal [6]. In addition, exercise might influence the midbrain periaqueductal grey (PAG) and the rostral ventromedial medulla (RVM) network by modulating pain transmission on a central level, which is associated with central endogenous release of inhibitory neurotransmitters in the central nervous system, such as GABA, leading to an inhibition of pain signaling [6,7,8].

Several studies were conducted with the aim to gain more precise insights into the physiological responses of EIH, and a wide range of research has been conducted since. When looking at the durations used in these exercise studies, it becomes evident that the majority of studies evaluated the EIH effects of rather short exercise sessions with exercise durations of 30 min or less [4]. Very few studies employed longer aerobic exercises to investigate the EIH phenomenon. This is somewhat surprising as one of the first studies reporting the phenomenon of EIH presented the case of a 40 min aerobic running episode at self-selected speed in which hypoalgesia was observed with reduced pain sensitivity after the run [9]. However, in another early study, 12 long distance runners conducting a 44 min run at an intensity of 85% of VO_2_max showed increased pain sensitivity, i.e., hyperalgesia, after the running exercise compared to pre-values [10]. 

Based on the present literature—particularly the paucity of longer duration studies—the present study seeks to evaluate the EIH effects in healthy participants employing three different aerobic exercise sessions using the same intensity but different durations, i.e., 30, 45, and 60 min, respectively. The gained knowledge might be particularly useful for concrete recommendations for aerobic physical activities and their potential pain-reducing effects.

## 2. Materials and Methods

### 2.1. Study Population and Experimental Design

An a priori power analysis using G*power (Version 3.1.9.4, Heinrich Heine University, Dusseldorf, Germany) for a repeated measures ANOVA was calculated beforehand and resulted in 32 subjects. A power of (1 − β) = 0.80, an α-error probability of 0.05, and a medium effect size of f = 33 were assumed. A dropout rate of 10% was further added resulting in the final sample size of 36 subjects.

Subjects’ characteristics are presented in Table 1. Subjects were included in the present study when fulfilling the following inclusion criteria: healthy, 18 to 35 years old, and a regular weekly physical activity of six or less hours, as it was shown that highly active athletes show an increased pressure pain tolerance [11]. Subjects were excluded if they reported chronic or acute pain, suffered from any orthopedic injuries, or were taking any analgesics regularly.

The study was designed as a quasi-randomized controlled crossover study and all participants conducted all three exercise sessions and one control session. An overview of the study design can be found in Figure 1.

This study was conducted in accordance with the principles of good clinical and ethical practice and the local ethic committee. Along with the Declaration of Helsinki, all participants gave written informed consent after being informed about the study protocol.

### 2.2. Pre-Experimental Test

All subjects visited the laboratory for a pre-experimental test. Here, a general health check-up was conducted by a clinician including anamnesis and electrocardiogram examination, as well as height and weight measurements. Further, subjects performed an incremental test exercise up to exhaustion, as conducted before [7], on a bicycle-ergometer (Excalibur Sport, Lode, Groningen, The Netherlands) (start 50 watts, increment 50 watts every three min) to determine the individuals’ relative maximal oxygen consumption (VO_2_max). The following criteria were used to check that all participants reached their VO_2_max: (1) plateau of VO_2_ achieved, (2) exceeding a calculated peak heart rate (200–age), (3) inability to maintain the effort, and (4) rate of elimination of CO_2_/O_2_ ratio of > 1.05 [12,13,14]. All participants fulfilled ≥3 of these criteria. Spirometric data were measured using the Vyntus CPX system (CareFusion, Hoechenberg, Germany). VO_2_max was recorded as the highest value of the measurement. Further, 20 µL blood was sampled from the earlobe for evaluation of the lactate concentration pre and post exercise and subsequently measured (Biosen S-Line Lab, EKF Diagnostics, Penarth, UK). In addition, this pre-experimental test was used as a familiarization session to avoid any confounding results due to the novelty of the measurement and to accustom the subjects with the pressure pain testing (PPT) and conditioned pain modulation (CPM) methods.

### 2.3. Exercise Sessions 

Following the pre-experimental test, subjects visited the laboratory for another three times to conduct the exercise and control sessions with at least seven days of pause between the test sessions to avoid any carry-over effects, as conducted before [15,16,17]. Subjects were asked to refrain from vigorous physical activity and not to take any analgesics 24 h prior the respective exercise session. After ten minutes of rest, the subjects’ PPT and CPM (pre) were assessed. Then, 20 µL blood was sampled from the earlobe for evaluation of the lactate concentration and subsequently measured. Subjects conducted the respective exercise session, which lasted for 30, 45, or 60 min. The intensity in terms of watt power used was set to 75% of the subjects’ VO_2_max in each exercise session as determined in the pre-experimental test (see Table 1). The watt power employed was constant throughout the exercise sessions. After finishing the respective exercise session, lactate was sampled, and PPT as well as CPM assessments were performed (post). Heart rate (HR) was protocolled before the exercise, then every ten minutes, and finally at the end of the exercise. Rate of perceived exertion (RPE) values were asked every ten minutes and at the end of the exercise. The order of the separate sessions was randomly allocated. Before the 30 min exercise session, a resting control session was included. Hence, after 10 min of rest, PPT and CPM assessments were conducted before and after subjects rested for 30 min. Subsequently, the 30 min session started as described above after another 10 min resting pause.

### 2.4. Pressure Pain Threshold Assessment

For the assessment of EIH, PPT were used as this method of pain sensitivity testing was reported to provide the most reliable results [5]. This was performed by using a handheld digital algometer (FPX 25 Compact Digital Algometer, Wagner Instruments, Greenwich, CT, USA). PPT measurements were conducted as described before to detect subjects’ pain sensitivity [18,19]. Pressure pain was applied to the elbow-, knee-, and ankle-joints bilaterally, as well as to the sternum and forehead via a one cm^2^ rubber tip. The applied pressure was increased with an increment rate of 10 newton (N) per second. Subjects were instructed to say “now” when they perceived the pressure stimulus painful for the first time. The peak applied force (N) was recorded. To prevent any tissue damage, a cut-off value of 140 N was determined beforehand, and for those participants reaching this pressure force, this cut-off value was used for later analysis [18]. The measurements were conducted three times with 10 s of pause in-between, and the mean value was used for analysis with lower values indicating enhanced pain sensitivity.

### 2.5. Conditioned Pain Modulation Assessment

The CPM test was employed with the aim to evaluate the subjects’ descending pain inhibition pathway [20]. The test procedure consisted of two parts. First (A), the individual test stimulus (stand-alone test stimulus; saTS) was applied using a heat stimulus, which was applied for 30 s via a thermode via a 9 cm^2^ contact area (TSA-II; controlled via Thermal Sensory Analyzer 2001 TSA-II, Medoc Ltd., Ramat Yishay, Israel) to the dominant forearm and subjects were asked to rate the pain intensity every 10 s on a NRS (0–100). The temperature of the test stimulus was calibrated at the day of the pre-experimental test. For this purpose, the thermode with an initial temperature of 32 °C was placed on the dominant forearm. After starting the test, the temperature increased by 1 °C per second. The heat to be applied was limited to a maximum heat value of 50 °C to avoid tissue damage. This test stimulus was to be classified by the subjects at a heat pain of 60 on a numerical rating scale (NRS) of 0–100. The software (TSA-II NeuroSensory Analyzer, version 6.1.19.4, Medoc Ltd., Ramat Yishay, Israel) indicated the applied temperature. This test was performed three times; the mean value was used for further analysis.

Part (B) of the procedure started after a restitution pause of one minute by applying the cold pain stimulus (stand-alone conditioning stimulus; saCS). The non-dominant arm was immersed into a circulating cold-water bath (7°; basin B-18, 18 L; thermoregulator TE-10D; immersion cooler, RU-200; Techne, Staffordshire, UK) for one minute. Subjects rated this standalone conditioning test stimulus intensity on a NRS (0–100) three times every 10 s. After 30 s, the test stimulus was applied in parallel for 30 s. Subjects were asked to rate the test stimulus under conditioning influence (ciTS) three times every 10 s on a NRS (0–100). After 30 s, the thermode temperature decreased back to the baseline temperature of 32 °C. The CPM response (CPMre) was calculated by subtracting the NRS value from the saTS from the NRS value from the ciTS. Hence, negative values suggest increased pain inhibition via CPM.

### 2.6. Statistics

Performance parameters observed during the three different exercise intervention were compared and presented descriptively in Table 2. In addition, RPE, HR, and lactate values measured at the end of each intervention were compared using a one-way ANOVA as data were normally distributed (using the Shapiro–Wilk test) and sphericity was checked (using the Mauchly test) and the Greenhouse–Geisser adjustment was used if necessary. To evaluate whether baseline values were different in terms of the PPT pre-values of each landmark, a one-way ANOVA with repeated measures was conducted across the four interventions. If a statistically significant difference was observed, a post hoc analysis with Bonferroni adjustment was calculated. These analyses were calculated for the whole participant sample and separately for female and male participants. To detect effects of the four interventions on PPT values of the different landmarks, a three-way ANOVA with the factors ‘intervention’ (control, 30, 45, and 60 min, respectively), ‘time point’ (pre, post), and ‘landmark’ (sternum, forehead, left and right elbows, left and right knees, and left and right ankles) was calculated. Further, the deltas of PPT (ΔPPT) were calculated by subtracting the pre-values from the post-values of the respective landmark and intervention. Subsequently, a two-way ANOVA was conducted for ΔPPT with ‘intervention’ and ‘landmark’ as factors. To detect effects of the four interventions in the CPM related parameters, i.e., saTS, saCS, ciTS, and CPMre, a two-way ANOVA was calculated with the factors ‘time point’ and ‘intervention’. It was further evaluated, how many participants responded to the three exercise sessions, i.e., whether EIH occurred on an individual level. Hence, participants were defined as responders if they exerted an increase in PPTs that was larger than the usual intra-individual variation in PPTs. To obtain a measure for variation in PPT, the standard error of measurement (SEM) of the four baseline PPT measurements was calculated for each landmark [21]. To so, the following formula was used SEM = $\mathrm{SD}\sqrt{1-r}$, with r being the intraclass correlation coefficient of the baseline values of each landmark [22].

Statistical analyses of the data were performed using the statistics software package SPSS 27 (IBM©, Armonk, NY, USA). Data are presented as means ± standard deviation. Differences were considered significant with *p* < 0.05.

## 3. Results

Of the initially included 36 subjects, 36 conducted all sessions of this study, and no dropout was recorded. Performance parameters of the pre-experimental test are presented in Table 1, and the performance-related data measured during the three different exercise interventions, i.e., RPE, HR, and lactate concentrations, are presented in Table 2. Evaluation of the PPT pre-values showed that no difference (*p* > 0.05) in pre-values was observed in any landmark between the exercise interventions for the entire subject sample as well as for female and male subjects, respectively. PPT values were not affected by any of the four interventions, and these results are illustrated in Figure 2. Further, ΔPPP were not affected by any of the interventions (*p* > 0.05; data not shown). The evaluation of responsive and non-responsive subjects revealed a low number of responders with a mean number of 12.0 (±2.1), 9.8 (±2.5), and 7.9 (±2.6) responders for the 30, 45, and 60 min session, respectively, depending on the landmarks observed. A detailed presentation of the responders can be found in Appendix A. Results regarding CPM related parameters, i.e., saCS, saTS, ciTS, and CPMre, are presented in Table 3.

## 4. Discussion

The aim of this study was to evaluate the acute exercise-induced hypoalgesic effects of three exercise interventions differing in the duration on PPT and CPM. Regarding PPT, results indicate that none of the interventions led to exercise-induced hypo- or hyperalgesia at the joints, nor at the sternum or the forehead. In addition, no change was observed in the CPM response. These results were not as expected, as the phenomenon of acute hypoalgesia following exercise is well described in the literature and most robust following aerobic exercise [2,3,5]. However, some studies also report that aerobic exercise does not lead to hypoalgesia following an acute aerobic exercise. Neither hypo- nor hyperalgesic effects were observed in the work by Hakansson et al., who evaluated the acute effect of a high-intensity interval (10 × 1 min intervals at 90% peak heart rate) and a moderate-intensity continuous (30 min at 65–75% peak heart rate) bicycle exercise on PPT [23]. Kodesh and Weissman-Fogel also report that neither a highly intensive interval exercise (4 min at 85% of the heart rate reserve (HRR) and 2 min of recovery at 60% of the HRR in each cycle, for a total of 30 min) nor a continuous exercise (70% of the heart rate reserve, for a total of 30 min) on a bicycle ergometer led to any changes in pain sensitivity using PPT measurements at the dominant hand [24]. Monnier-Benoit and Groslambert also observed that a 30 min steady-state cycling exercise at 75% of the VO_2_max led to no change in a pressure pain stimulation test in which the participants had to rate a constant pressure pain stimulus on a numeric rating scale (NRS) applied on the index finger for 120 s. These results were observed in trained cyclists and untrained men [25]. Our results show that no divergent effects between the different exercise interventions occurred when comparing 30, 45, and 60 min exercises at the same intensity. Evidently, only little research has been conducted to evaluate the effects in EIH of even longer duration studies (i.e., >30 min) [4]. Yet, in contrast to these longer duration studies, shorter duration studies which go along with a higher intensity seem to result in EIH more robustly, and we showed in a recent study that EIH is induced more prominently the higher the intensity is [7]. In this study, participants performed four 30 min bicycle exercises at four different intensities, i.e., 60%, 80%, 100%, and 110% of the individual anaerobic threshold (IAT). The results indicate a main time effect with lower PPT values measured post exercise. Yet, no interaction effects (time X intensity) were observed. However, hypoalgesia was observed more prominently after the 100% and 110% exercise sessions. The intensities used in this study can only be compared to a limited degree to the intensity used in the present study, as the intensity was defined differently (relative to the individuals VO_2_max and relative to the IAS). However, when comparing the employed absolute and relative power in the study by Tomschi et al. (2022) [7], it can be concluded that the 80% IAS session with 164 ± 51 W (2.2 W/kg) resembles the study presented herein (absolute power 172 ± 39 W; relative power 2.3 W/kg) in terms of intensity the most. Looking at the post hoc calculations of the beforementioned study, it can be observed that hypoalgesia occurred after the 80% IAT session only at the ankle joints and no other landmark, which were the same as in the present study. In contrast, the 100% and 110% IAT sessions resulted in a more global EIH, which was observed also at the knees, elbows, and the sternum. This observation implies that a higher intensity, or the combination of a higher intensity and duration, might be more important for determining the degree of EIH after aerobic exercise than the duration alone [4]. 

Early research proposed that there might be certain thresholds for intensity (>50% VO_2_max) and duration (>10 min) for aerobic exercises to induce acute hypoalgesia [26]. For instance, Naugle et al. showed that 20 min of vigorous exercise at 70% of HR reserve increased PPT whereas PPT was unaltered after moderate exercise at 50% of HR reserve [27]. Yet, a specific threshold for duration as well as intensity needed to induce EIH robustly is still to be determined and our results question this concept of one universal threshold. The intensity used in the present study was set to 75% of the VO_2_max. Looking at the corresponding RPE and HR values, it becomes evident that the intensity used was perceived as moderately hard; yet, HR values were in a moderate range with an HR ranging between 162.5, after the 30 min intervention, and 165.8 or 167.8 beats/min, after the 45 and 60 min intervention, respectively, especially when considering the participants’ young age of approximately 26 years. Further, lactate concentrations measured at the end of each interventions show that participants most likely utilized energy via aerobic and only in part anaerobic pathways as the lactate concentrations ranged between 4.2 (30 min intervention), 3.4 (45 min intervention), and 3.0 mmoL/L (60 min intervention), respectively. Therefore, the intensity employed was most likely slightly too low to induce hypoalgesic effects, as the participants did not reach the anaerobic threshold. A study by Vaegter at al. revealed that 15 min of cycling at the anaerobic lactate threshold induced hypoalgesic effects measured using PPT at the quadriceps muscles [28].

Another theory has been brought forward suggesting that higher-intensity exercises going along with muscular discomfort or pain lead to a greater EIH magnitude than non-painful exercises [25]. As observed in the HR, RPE, and lactate values, the intensity used in the present study does not correspond to the exercise discomfort or the muscular pain observed during highly intensive exercise, and hence, changes in pain perception of large magnitudes were not to be expected. This theory goes along with the suggestion that a certain exercise intensity needs to be reached to induce hypoalgesic effects. An increase in β-endorphin is suggested to result from severe exercise. β-endorphin possesses a morphine-like activity and increases hypoalgesic effects [29]. It needs to be considered that β-endorphin levels were only starting to elevate in an incremental exercise test when the anaerobic threshold was exceeded [30]. More particularly, during this incremental cycling ergometer test (starting at 50 watts, with an increase of 50 watts every three minutes), β-endorphin was slightly elevated (not significantly) when exceeding the anaerobic threshold and a lactate concentration of 4 mmoL/L. β-endorphin was only significantly elevated when reaching 300 watts, corresponding to 18 min of cycling, and a lactate concentration roughly between 7 and 10 mmoL/L. Hence, it was suggested that the extent of anaerobic energy used might correlate with the β-endorphin production, which was also supported by another study showing that moderate endurance steady state exercises do not lead to an increase in endorphin production [31]. Peripherally, β-endorphins induce pain reducing effects by binding to opioid receptors at pre- and post-synaptic nerve terminals. This leads to an inhibition of substance P, which is known to modulate pain transmission. Furthermore, β-endorphins induce analgesia on central nervous system level by increasing dopamine release via GABA release inhibition [32]. The results observed in the present study might lead to the conclusion that the used intensity was not sufficient to induce an increase in β-endorphin levels, which in turn would have led to anti-nociceptive activity as the observed lactate levels only slightly exceed the 4 mmol/l lactate threshold and were most likely steady throughout the duration of the respective intervention.

The landmarks used for PPT measurements in the present study need to be considered since PPT were measured at the elbow, knee, and ankle joints, more precisely in the respective joint space. Many other studies demonstrating EIH employed different landmarks for PPT measurements, e.g., the rectus femoris and tibialis anterior muscles [33], quadriceps and upper trapezius muscle [28], or the rectus femoris and brachioradialis muscles [6]. Due to the different body structures used, one might argue that the pressure pain thresholds might be different in different underlying tissues due to, e.g., different nociceptive and tactile sensitivity and/or divergent peripheral afferent nerve fibers, as well as different innervation densities at the joints compared to the musculature. In bony structures at the joints, these fibers are mostly present in the periosteum rendering this structure most sensitive to nociceptive stimuli. A-delta fibers and C-fibers are activated upon mechanical distortion, such as the PPT procedure [34]. Indeed, the same experimental pain stimulation was perceived as more painful when being applied to muscle structures compared to bony structures [35]. To the best of our knowledge, only one study, besides the aforementioned study [7], evaluated EIH by measuring PPT at the joints. Krüger et al. showed that an incremental bicycle ergometer-test up to exhaustion did not lead to hypo- or hyperalgesia at the joints. These results are in line with the present study. Contrastingly, Krüger et al. further report hyperalgesic effects observed at the sternum and forehead [19]. 

Despite the proposed explanations, the exact reasons for the lack of observed EIH in the present study remain fairly elusive. One might also speculate that the lack of effects observed can also be attributed to the non-responsiveness of subjects recruited, as a large proportion of participants of this study did not respond in a hypoalgesic manner to the exercises conducted (see Appendix A). Even though EIH is believed to be rather robust on a group level, the individual response to exercise is more variable [4]. Studies have investigated the stability of EIH in response to different exercise modes, including aerobic exercise, across different days and results show that some individuals consistently show hypoalgesia (responder). Yet, some individuals constantly show no change or hyperalgesia (non-responder) after exercise, but some individuals also presented both hypoalgesic and hyperalgesic responses between several testing days [21,36,37]. 

The present study also evaluated the effects of the respective exercise sessions on CPM. Results show that the different exercise durations do not lead to changes in the CPMre. Yet, we observed a main effect on the saTS with lower values being observed after the sessions. Interestingly, no interaction effects were observed. However, post hoc tests revealed lower values after the 45 and 60 min intervention, respectively, indicating reduced pain sensitivity with respect to heat stimuli after these exercise sessions, which was also observed in a previous study [38]. Further, some influencing factors need to be discussed in this context when evaluating the CPM response. First, the exercises most likely led to an increase in body temperature, which might have blurred the CPM results because for the test and conditioned stimuli thermal stimuli were employed. Second, sensitivity of CPM might be limited to yield statistically significant differences within the recruited subject sample of this study as healthy and young subjects usually reach maximum values [39], which is also observed in our study.

The major strength of the present study is that three different exercise sessions of different durations and one passive control session were evaluated with respect to the EIH phenomenon in a relevant large number of participants. Yet, there are also some limitations that need to be discussed. First, the methodical setup of this study resulted in no change in pain perception. These results were not as expected, but at the same time, they provide some valuable insights into the mechanisms of EIH. Second, the study was designed as a quasi-randomized study, and the control session was followed by the 30 min exercise session at the same day. Yet, the results reveal that the pre-values of both interventions were not different, indicating that the outcomes were not influenced by this setup. Third, only healthy young participants were included, and the results are therefore not directly transferable to older subjects and especially not to any clinical populations. Forth, the menstrual cycle phase of female subjects might affect pain sensitivity, but research shows that the menstrual cycle phase has little effects on the perception of pain in healthy, pain-free women [40], which is also supported by our study, as no differences were observed between baseline values.

## 5. Conclusions

This study aimed to evaluate the effects of differently long bicycle exercise sessions conducted at the same intensity on exercise-induced hypoalgesia measured by PPT at the joints, sternum, and forehead. The results indicate that neither hypo- nor hyperalgesic effects occur, indicating that EIH might not be as stable as is believed. In addition, no effects were observed regarding CPMre. Yet, these results were unexpected and might, to a certain degree, be explained by too low intensities, the landmarks chosen for PPT measurements, or non-responsive subjects. 

## Figures and Tables

**Figure 1 biology-12-00222-f001:**
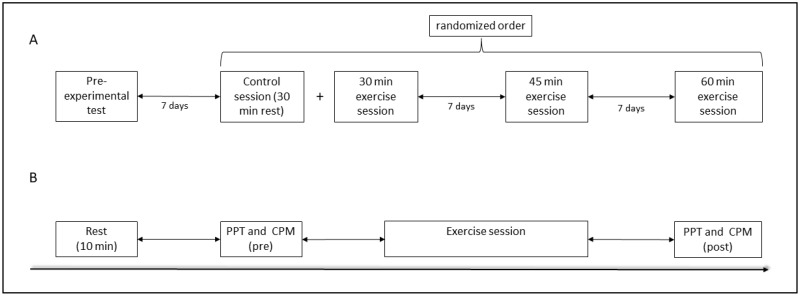
General study design in which each participant visited the laboratory for a total of four times (**A**); experimental design of one exercise session including PPT and CPM measurements (**B**). PPT = Pressure pain thresholds; CPM = Conditioned pain modulation.

**Figure 2 biology-12-00222-f002:**
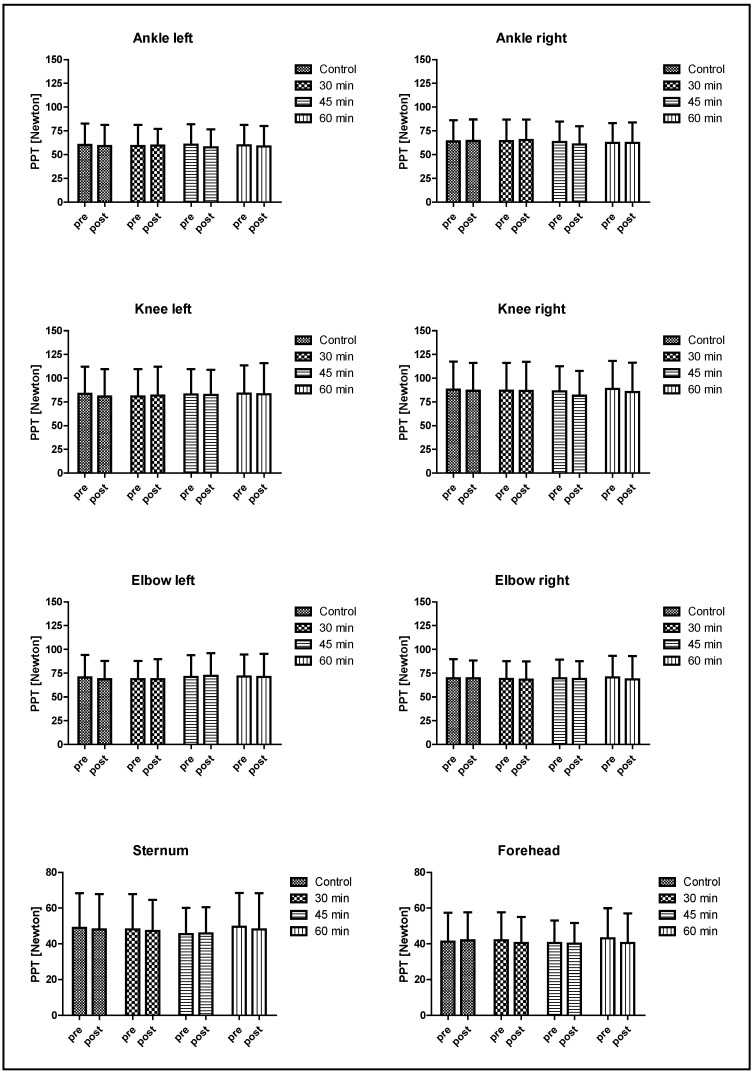
PPT values measured pre and post exercise. No significant main or interaction effects were observed. Data are presented as means ± standard deviation.

**Table 1 biology-12-00222-t001:** Anthropometric and performance data obtained in the pre-experimental test of participating subjects. Data are presented as means ± standard deviation. VO_2_max = Relative maximal oxygen consumption; HR = Heart rate.

Parameter	Subjects (*n* = 36)
Gender [male/female]	18/18
Age [years]	26.6 ± 3.2
Height [m]	1.8 ± 0.1
Weight [kg]	73.3 ± 11.1
VO_2_max [mL/min/kg]	42.5 ± 8.0
75% of VO_2_max [mL/min/kg]	31.8 ± 6.0
Max. heart rate [1/min]	184.2 ± 9.0
Peak Power [W]	259.7 ± 57.1
Max. lactate [mmol/L]	11.3 ± 2.3
Power at 75% of VO_2_max [W]	172.2 ± 39.1

**Table 2 biology-12-00222-t002:** Rate of perceived exertion (RPE), heart rate (HR), and lactate concentrations measured during the three different exercise interventions. * indicates significant difference compared to RPE, HR, and lactate, respectively, at the end of the 30 min intervention. Data are presented as means ± standard deviation.

Parameter	Time	30 Min Exercise Session	45 Min Exercise Session	60 Min Exercise Session
RPE (6–20)	10 Min	13.4 ± 1.8	12.9 ± 1.6	12.9 ± 1.7
	20 Min	14.8 ± 1.8	14.5 ± 1.5	14.2 ± 1.6
	30 Min	15.6 ± 2.0	15.2 ± 1.6	15.2 ± 1.7
	40 Min	X	16.1 ± 1.7	15.7 ± 1.7
	45 Min	X	16.5 ± 1.7 *	X
	50 Min	X	X	16.3 ± 1.8
	60 Min	X	X	16.7 ± 1.9 *
HR (1/Min)	0 Min	64.5 ± 11.0	68.3 ± 11.8	68.8 ± 14.2
	10 Min	151.2 ± 13.9	150.2 ± 14.3	151.4 ± 13.6
	20 Min	159.2 ± 13.2	157.9 ± 13.2	158.4 ± 10.8
	30 Min	162.5 ± 12.9	162.2 ± 12.5	163.1 ± 9.3
	40 Min	X	164.8 ± 12.6	164.8 ± 9.3
	45 Min	X	165.8 ± 12.4 *	X
	50 Min	X	X	166.5 ± 8.8
	60 Min	X	X	167.8 ± 9.4 *
Lactate (mmoL/L)	Pre	0.9 ± 0.3	0.9 ± 0.2	1.0 ± 0.3
	Post	4.2 ± 1.9	3.4 ± 1.5 *	3.0 ± 1.7 *

**Table 3 biology-12-00222-t003:** CPM-related data before and after the respective exercise and control sessions. * indicates a significant difference vs. before the same session. # indicates a significant difference vs. before the 30 min exercise session. Data are presented as means (standard deviation).

	Time Point	Control Session	30 Min Exercise Session	45 Min Exercise Session	60 Min Exercise Session
Stand-alone conditioning test stimulus (saCS) (NRS points)	pre post	57.6 (19.6) # 59.6 (20.0) *	59.6 (20.1) 55.0 (19.4) *	57.6 (21.3) 58.0 (22.0)	56.1 (21.4) 57.6 (20.6)
Stand-alone test stimulus (saTS) (NRS points)	pre post	52.4 (12.1) 52.1 (11.7)	52.0 (12.3) 47.2 (13.3)	52.5 (14.0) 46.4 (15.0) *	52.8 (12.3) 46.0 (13.7) *
Test stimulus Intensity under conditioning influence (ciTS) (NRS points)	pre post	27.2 (18.0) 27.5 (19.1)	27.6 (19.3) 23.9 (17.9)	26.6 (17.1) 23.8 (16.7)	29.7 (18.2) 22.9 (17.2)
CPM response (CPMre) (NRS points)	pre post	−25.2 (17.8) −24.6 (16.1)	−24.4 (16.9) −23.3 (16.4)	−26.0 (17.5) −22.6 (16.7)	−23.1 (16.9) −23.1 (17.8)

## Data Availability

The data presented in this study are available on request from the corresponding author.

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
