# Peer review of "Does Exercise-Induced Hypoalgesia Depend on Exercise Duration?"

_biology, 2023, doi:10.3390/biology12020222_

Round 1

Reviewer 1 Report

Although this article appears in a new journal, I have already reviewed this revised version in the International Journal of Environmental Research and Public Health. Since the authors did not change anything in the version presented earlier, there is no need for me to change the review written earlier:

I read with interest the responses of the authors of the article to my previous comments, as well as to the comments of the second reviewer. I would like to make a few comments. The first reviewer believes that the article cannot be accepted not only because the results of the study are negative, but also because the reviewer does not see novelty in the study. Unfortunately, due to the brevity of the review, the validity of such a view cannot be assessed, it remains to take his word for it. However, the authors' response to the reviewer is incomplete.

In my own review, I tried to encourage the authors not only to state the facts obtained, but also to explain why the results turned out to be the way they were. There is a feeling that partly such results were obtained due to design features. Therefore, we have to continue the discussion.

1. The authors did not answer the question - why, to calculate the maximum load during the initial test, they used not the traditional formula for calculating heart rate (220-age), but another one (200-age).

2. Also, the authors did not answer the question - why, during the initial test, the average maximum heart rate was 184 beats per minute (if you use the authors' formula, it should be 174 beats per minute).

3. Thanks to the authors for explaining the subtle differences between trial termination criteria and VO2max achievement criteria. A natural question arises - what were the criteria for achieving VO2max in the study cohort? As far as I understand, the trial was terminated due to subjective exhaustion, which meets the criterion (3) of achieving VO2max according to the authors (inability to maintain the effort). So, did the subjects meet the other three VO2max criteria? Or did the subjects present 2 or more criteria? This question follows from the previous one. In addition, one more question - in case of reaching the maximum heart rate (200-age), the test was continued or stopped (because it seemed like the criterion for achieving VO2max was reached).

4. The answer to my question 2 is inaccurate - the authors did not understand the question. It was not about the method of dosing the load during training, but about conducting the graded pre-experimental test. Therefore, all the arguments about the dosing of the load in watts during exercise sessions do not relate to the essence of my question.

5. The answer to my comment 3 did not satisfy me either. I do not see a fundamental difference in the methods of dosing the submaximal load (either from the maximum aerobic capacity achieved, or from the load in watts achieved at the same time). However, the method of determining the maximum aerobic productivity is of fundamental importance, but the authors diligently avoid discussing this issue.

Author Response

Reviewer 1:

Although this article appears in a new journal, I have already reviewed this revised version in the International Journal of Environmental Research and Public Health. Since the authors did not change anything in the version presented earlier, there is no need for me to change the review written earlier:

I read with interest the responses of the authors of the article to my previous comments, as well as to the comments of the second reviewer. I would like to make a few comments. The first reviewer believes that the article cannot be accepted not only because the results of the study are negative, but also because the reviewer does not see novelty in the study. Unfortunately, due to the brevity of the review, the validity of such a view cannot be assessed, it remains to take his word for it. However, the authors' response to the reviewer is incomplete.

In my own review, I tried to encourage the authors not only to state the facts obtained, but also to explain why the results turned out to be the way they were. There is a feeling that partly such results were obtained due to design features. Therefore, we have to continue the discussion.

Comment 1. The authors did not answer the question - why, to calculate the maximum load during the initial test, they used not the traditional formula for calculating heart rate (220-age), but another one (200-age).

Answer 1: Thank you for this advice and we will explain the choice of this formula in the pre-experimental test. As we chose to employ certain criteria to check whether the participants reached their VO2max, we also included a heart rate formula to do so. As some young and rather active men and women are physiologically not able to reach classically calculated maximum heart rates during maximum exhaustion (when using for example 220-age), we chose to use a more conservative formula with 200-age. This decision was made with the aim to not exclude otherwise suitable participants. We would like to illustrate this with the following example based on one large study investigating HRmax values in recreationally active men and women using an incremental bicycle ergometer test [1]. In this study, which included 128 recreationally active participants (age: MEAN: 31.5; SD: 16.7), subjects reached a mean HRmax of 179.5 (SD: 20.2). The calculated HRmax using the formula 220-age would have therefore been approximately 188.5 bpm. Hence, several participants did not reach or exceed the calculated HRmax. When also considering the SD, several participants in this respective study reached even lower HRmax values by at the same time performing until subjective exhaustion. Yet, when using the formula 200-age, the vast majority participants would have been above the calculated HR. This might be one reason why some studies use the formula 200-age for bicycle interventions instead of 220-age [e.g., 2, 3]. We added the respective references into the manuscript in line 108.

Comment 2. Also, the authors did not answer the question - why, during the initial test, the average maximum heart rate was 184 beats per minute (if you use the authors' formula, it should be 174 beats per minute).

Answer 2: Thank you for this hint and we must agree that we stated the respective sentence in a slightly misleading manner. We would like to stress that this was one of four criteria to check for maximum exhaustion and hence, for the validity of the VO2max. Besides, 200-age was the lowest limit, which had to be exceeded by the participants and we adjusted the particular sentence to describe this more clearly in line 107. This means that all pre-experimental tests were only terminated due the participants’ exhaustion and not when participants reached the calculated lower limit of their respective maximum heart rate. This is why the stated mean maximum heart rate in Table 1 is with 184.2 beats/min higher than 173.4 beats/min, as calculated via 200-26.6, which is the mean age of the sample.

Comment 3. Thanks to the authors for explaining the subtle differences between trial termination criteria and VO2max achievement criteria. A natural question arises - what were the criteria for achieving VO2max in the study cohort? As far as I understand, the trial was terminated due to subjective exhaustion, which meets the criterion (3) of achieving VO2max according to the authors (inability to maintain the effort). So, did the subjects meet the other three VO2max criteria? Or did the subjects present 2 or more criteria? This question follows from the previous one. In addition, one more question - in case of reaching the maximum heart rate (200-age), the test was continued or stopped (because it seemed like the criterion for achieving VO2max was reached).

Answer 3: Thank you for this hint and we are happy to provide more information regarding this issue. All participants needed to fulfill ≥3 of the four criteria, which was achieved by all the participants. We added this piece of information into the methods section (see lines 108f)

Comment 4. The answer to my question 2 is inaccurate - the authors did not understand the question. It was not about the method of dosing the load during training, but about conducting the graded pre-experimental test. Therefore, all the arguments about the dosing of the load in watts during exercise sessions do not relate to the essence of my question.

Answer 4: We are sorry for this little misunderstanding, and we hope that we clarified this issue with the previous answer to comment 3 of this review.

Comment 5: The answer to my comment 3 did not satisfy me either.

Previous comment 3: Considering the explanations given by the authors on comparing their data with previous work, it becomes clear that the authors initially used a less intense load, which did not cause EIH for a short duration. Therefore, a more correct hypothesis of the study should have been formulated as follows: can non-long-term, non-intense physical activity ineffective in inducing EIH achieve EIH with an increase in duration?

Previous answer 3: We are happy to respond to this comment. In our previous study published in the European Journal of Applied Physiology, the higher intensity sessions led to a more global hypoalgesic effect compared to the lower intensity sessions. Yet, all sessions (also the low intensity sessions) led to hypoalgesia at the ankles. Therefore, in the present paper, we also provided further possible reasons in the discussion section why no hypoalgesia was observed in the present study. Regarding your suggested hypotheses: We did not provide any hypotheses in the present paper. Yet, if we understand your suggestion correctly, one might think about whether an increase in the exercise duration, by at the same time remaining a low intensity, results in more pronounced hypoalgesia. However, as described in lines 284f, the intensities of the two studies cannot directly be compared to each other as the former study used certain percentages of the individuals’ anaerobic thresholds to calculate the intensity (watt power). Contrastingly, the present study uses 75% the individuals’ VO2max to calculate the intensity (watt power), which is why we would like to abstain from such an argumentative line.

I do not see a fundamental difference in the methods of dosing the submaximal load (either from the maximum aerobic capacity achieved, or from the load in watts achieved at the same time). However, the method of determining the maximum aerobic productivity is of fundamental importance, but the authors diligently avoid discussing this issue.

Answer 5:

Thank you for this remark and we are happy to provide more information regarding the pre-experimental test and how the maximum aerobic capacity was determined. In both studies, i.e., the study presented herein and our previous study [4], the same procedure was used to determine the maximum aerobic capacity. This test used in both studies consisted of an incremental bicycle-ergometer test up to exhaustion as described in lines 101ff in the manuscript and in the previously published paper: “The IAT was determined using an incremental stepwise protocol (start: 50 watts, increment: 50 watts every 3 min) on a bicycle ergometer (Excalibur Sport, Lode, Groningen, Netherlands) until exhaustion” [4]. We adjusted this sentence in the present manuscript and added this reference (see lines 102f). Furthermore, we agree that the intensity chosen might have been slightly too low to induce hypoalgesia, but we are still of the opinion that our added paragraph in lines 259-279 discusses this issue in an adequate manner as the absolute and relative power was similar in the previously published study and the study presented herein (164 vs. 172 W and 2.2 W/kg vs. 2.3 W/kg). We added this piece of information into the manuscript (see lines 269ff). Besides, we acknowledge the issue of the slightly too low intensity in lines 280-325, especially in lines 295f where we wrote “Therefore, the intensity employed was most likely slightly too low to induce hypoalgesic effects”.

References used in this response letter to reviewer I:

[1] Roecker, K., H. Striegel, and H-H. Dickhuth. "Heart-rate recommendations: transfer between running and cycling exercise?." International journal of sports medicine 24.03 (2003): 173-178.

[2] Lach, Jacek, et al. "How to calculate a maximum heart rate correctly?." Folia Cardiologica 17.5 (2022): 289-292.

[3] Angerer, Peter, et al. "Comparison of cardiocirculatory and thermal strain of male firefighters during fire suppression to exercise stress test and aerobic exercise testing." The American journal of cardiology 102.11 (2008): 1551-1556.

[4] Tomschi, Fabian, Dennis Lieverkus, and Thomas Hilberg. "Exercise-induced hypoalgesia (EIH) in response to different exercise intensities." European Journal of Applied Physiology 122.10 (2022): 2213-2222.

Reviewer 2 Report

The present study explored the effects of exercise duration on EIH effects in healthy volunteers. The findings are somewhat surprising, as no EIH effects were observed. Overall, the study is well-conducted. However, there are a number of issues that could be addressed to further strengthen the manuscript.

Introduction

There are several typographical errors throughout the text. Please ensure to proofread and correct before resubmission.

The phrase “Physical activity is well known to reduce general and unspecific pain” is vague – Could the authors please be more specific (and include additional references) regarding the effects of physical activity on pain.

The phrase “Aerobic exercises, dynamic resistance exercises, as well as isometric exercises were shown increase pain thresholds and, thus reduce pain sensitivity of up 34 to 20% compared to a resting control condition” seems to be referring to a specific study, but the reader is not orientated as such. Please reword or specifically introduce the relevant paper.

Please include information regarding the mechanisms underlying EIH in the introduction, e.g.:

Wewege, M. A., & Jones, M. D. (2021). Exercise-induced hypoalgesia in healthy individuals and people with chronic musculoskeletal pain: a systematic review and meta-analysis. The journal of pain22(1), 21-31.

Borovskis, J., Cavaleri, R., Blackstock, F., & Summers, S. J. (2021). Transcranial Direct Current Stimulation Accelerates The Onset of Exercise-Induced Hypoalgesia: A Randomized Controlled Study. The journal of pain22(3), 263–274. https://doi.org/10.1016/j.jpain.2020.08.004

Rice, D., Nijs, J., Kosek, E., Wideman, T., Hasenbring, M. I., Koltyn, K., ... & Polli, A. (2019). Exercise-induced hypoalgesia in pain-free and chronic pain populations: state of the art and future directions. The Journal of Pain20(11), 1249-1266.

Method:

Please provide more detail as to how PPTs were collected. Was the average of three recordings used for analysis?

 Discussion

Further information is required as to why your findings (particularly the lack of EIH observed) may differ from those of existing research. Given the aim of the research, the lack of induced EIH should also be mentioned in the limitations section.

Author Response

Reviewer 2:

The present study explored the effects of exercise duration on EIH effects in healthy volunteers. The findings are somewhat surprising, as no EIH effects were observed. Overall, the study is well-conducted. However, there are a number of issues that could be addressed to further strengthen the manuscript.

Introduction

Comment 1: There are several typographical errors throughout the text. Please ensure to proofread and correct before resubmission.

Answer 1: Thank you for this hint and we re-read the manuscript accordingly and hopefully solved this issue.

Comment 2: The phrase “Physical activity is well known to reduce general and unspecific pain” is vague – Could the authors please be more specific (and include additional references) regarding the effects of physical activity on pain.

Answer 2: We are glad to adjust this phrase and we reformulated this sentence and focused more on the hypoalgesic effects of exercise and provided more references additionally dealing with chronic pain populations (see lines 36f and 40f).

Comment 3: The phrase “Aerobic exercises, dynamic resistance exercises, as well as isometric exercises were shown increase pain thresholds and, thus reduce pain sensitivity of up 34 to 20% compared to a resting control condition” seems to be referring to a specific study, but the reader is not orientated as such. Please reword or specifically introduce the relevant paper.

Answer 3: You are right, this statement is quite unspecific in terms of the type of exercise. Yet, the exact percentage is taken from the review by Vaegter and Jones [1]. They refer to two studies, which we have happily included into our introduction (see line 45). With this sentence we would simply illustrate to the reader what magnitudes of reduced pain sensitivity can occur after exercise. A more detailed comparison of existing studies relevant to our study and results, their methods used, as well as the results obtained is provided in the discussion section.

Comment 4: Please include information regarding the mechanisms underlying EIH in the introduction, e.g.:

Wewege, M. A., & Jones, M. D. (2021). Exercise-induced hypoalgesia in healthy individuals and people with chronic musculoskeletal pain: a systematic review and meta-analysis. The journal of pain, 22(1), 21-31.

Borovskis, J., Cavaleri, R., Blackstock, F., & Summers, S. J. (2021). Transcranial Direct Current Stimulation Accelerates The Onset of Exercise-Induced Hypoalgesia: A Randomized Controlled Study. The journal of pain, 22(3), 263–274. https://doi.org/10.1016/j.jpain.2020.08.004 [Titel anhand dieser DOI in Citavi-Projekt übernehmen]

Rice, D., Nijs, J., Kosek, E., Wideman, T., Hasenbring, M. I., Koltyn, K., ... & Polli, A. (2019). Exercise-induced hypoalgesia in pain-free and chronic pain populations: state of the art and future directions. The Journal of Pain, 20(11), 1249-1266.

Answer 4: We are happy to provide a short overview into the mechanisms underlying EIH in the introduction (see lines 48ff).

Method:

Comment 5: Please provide more detail as to how PPTs were collected. Was the average of three recordings used for analysis?

Answer 5: Yes, you are right. The average value of three measurements were used for analyses and were conducted with a pause of 10 seconds in-between the measurements. We described this procedure in lines 151f. We also added details of the measurements to further illustrate the measurement (see lines 151ff).

Discussion

Comment 6: Further information is required as to why your findings (particularly the lack of EIH observed) may differ from those of existing research. Given the aim of the research, the lack of induced EIH should also be mentioned in the limitations section.

Answer 6: Thank you for this comment and we were also rather surprised of the results obtained. We added one sentence to the limitations section (see lines 376ff). Yet, we are not the first authors to observe that exercise does not always lead to hypoalgesia acutely (see lines 241-255), even when a similar intensity was used (see the study by Monnier-Benoit and Groslambert (2006) and lines 250ff). In the discussion, we tried to discuss these “nil-findings” based on the methodological setup and proposed that these findings might be explained by the fact that the intensity was slightly too low to induce hypoalgesia, which can be interpreted using the lactate, HR, and RPE outcomes (see Table 2). In this context we also argued that the exercises conducted did not induce much or sufficient muscular discomfort or pain, which is also discussed to be partly responsible for EIH (see lines 299-323). Besides, we chose joint and reference landmarks to measure PPT as done before by our group. Using these landmarks, pressure is applied to bony structures. These landmarks might need a more severe exercise input in terms of intensity compared to muscular tissues to respond in a hypoalgesic manner (see lines 232-342) and a previous study of our group using the same landmarks as the study presented herein showed that higher intensity exercise goes along with a more global and widespread hypoalgesic response (see lines 258-277). Lastly, we argue that some participants might simply be seen as “non-responders”, which is also discussed in the previous literature (see lines 343-353). After a vast literature screening, we believe that these explanations cover the most striking explanations for the “nil-findings” of this study and no further changes were made in the manuscript. Yet, we are happy to add more aspects when suggested by the reviewers to discuss our results.

References used in this response letter to review II:

[1] Vaegter, Henrik Bjarke, and Matthew David Jones. "Exercise-induced hypoalgesia after acute and regular exercise: experimental and clinical manifestations and possible mechanisms in individuals with and without pain." Pain Reports 5.5 (2020).

Reviewer 3 Report

Dear editor, dear authors,

Thank you so much indeed for sending this manuscript to be reviewed by me.

It is considered an original, interesting paper that provides relevant information regarding EIH depending on exercise duration/

Although a very interesting objective but the study design chosen for this study looks really unbalanced and un-appropriate in order to achieve the aims of this study!

The first significant critics of mine goes for this unbalanced design: 36 participants were assigned in to three intervention group with different duration but the same intensity!!! In this case the total work performed by each of the three groups would be completely different so how could you say that the different reaction in different groups (in case there was any) is due to the different duration of the training and not the amount of the work????

The second critics is about the intensity chosen for all the three intervention groups: a: based on your inclusion criteria you recruited “36 Healthy, 18 to 35 years old, and a regular weekly physical activity of six or less hours, as it was shown that highly active athletes show an increased pain pressure tolerance”.

Do you think the intensity you chose for this study as 75 % vo2max is enough to stimulate the EIH in this active participants? I do not think so at all!!!

Is the 7 days selected for the washout, an enough time to avoid the carry over effects??? Is there any reference supporting your logic to choose 7 days???

Is only order randomization enough to avoid carry over effect as for four times of ppt assessment between the first control session and then the third day? What about avoiding the effect of two times PPT assessment in day one??? For sure there would be some effect of two times (pre test and post test) at the control session on the PPT assessments (pre test and post test) of the 30 min exercise session??? Don’t you think???

Considering the upper mentioned critics, I am afraid to say that I suppose this study has lots of methodological limitations so that making it inappropriate to be considered for publication in Biology (ISSN 2079-7737).

Best regards,

Author Response

Reviewer 3:

Dear editor, dear authors,

Thank you so much indeed for sending this manuscript to be reviewed by me.

It is considered an original, interesting paper that provides relevant information regarding EIH depending on exercise duration/

Although a very interesting objective but the study design chosen for this study looks really unbalanced and un-appropriate in order to achieve the aims of this study!

Comment 1: The first significant critics of mine goes for this unbalanced design: 36 participants were assigned in to three intervention group with different duration but the same intensity!!! In this case the total work performed by each of the three groups would be completely different so how could you say that the different reaction in different groups (in case there was any) is due to the different duration of the training and not the amount of the work????

Answer 1: Dear reviewer. Thank you for this comment and we are happy to explain the choice of the three interventions performed. First, this study was designed as a cross-over-study, meaning that every participant conducted all three interventions. We added this piece of information into the methods section of the study design (lee lines 75f). Of course, you are right: Longer exercise sessions go along with a higher total work performance and metabolic demand. Yet, this was exactly the initial aim of the study, i.e., to explore whether the duration of the exercise influences pain perception (see lines 70ff).

Comment 2: The second critics is about the intensity chosen for all the three intervention groups: a: based on your inclusion criteria you recruited “36 Healthy, 18 to 35 years old, and a regular weekly physical activity of six or less hours, as it was shown that highly active athletes show an increased pain pressure tolerance”.

Do you think the intensity you chose for this study as 75 % vo2max is enough to stimulate the EIH in this active participants? I do not think so at all!!!

Answer 2: Thank you for this thought. When considering the current literature regarding EIH, it is generally believed that an intensity of 75% of the individual’s VO2max is the minimum of inducing EIH (see the most cited study by Hoffman et al. [1] and the following reviews [2, 3]). When looking at the resulting heart rate (HR), lactate, and rate of perceived exertion (RPE) values (Table 2), it becomes evident that the intensity was in a moderate range, which is also discussed in the paper (see lines: 282ff). Based on these objective criteria reflecting the participants’ exertion and effort, we are of the opinion that the intensity was low, but not too low. Yet, this was the objective of this investigation as we tried to minimize the effect of a high or higher intensity on the outcomes as this study aimed to explore the effects of the different exercise durations on EIH. We also discussed this issue in lines 299-323.

Comment 3: Is the 7 days selected for the washout, an enough time to avoid the carry over effects??? Is there any reference supporting your logic to choose 7 days???

Answer 3: To the best of our knowledge, no study investigated whether 7 days of washout are enough to avoid carry-over effect. Yet, we would like to bring forward some assumptions on which we based our decision to use this particular washout period: EIH is believed to be an acute phenomenon “lasting ≤ 30 minutes after a single bout of exercise” [2]. Hence, possible carry-over effects are unlikely to potentially affect the next experimental session when separated by seven days. Therefore, this time frame is usually used in studies dealing with EIH [e.g., 4, 5, 6]. We added this piece of information into the methods section (see line 117).

Comment 4: Is only order randomization enough to avoid carry over effect as for four times of ppt assessment between the first control session and then the third day? What about avoiding the effect of two times PPT assessment in day one??? For sure there would be some effect of two times (pre test and post test) at the control session on the PPT assessments (pre test and post test) of the 30 min exercise session??? Don’t you think???

Answer 4: Thank you for that comment and we tried to evaluate this issue, i.e., whether the sequence/order of PPT measurements affected to outcome. This might have occurred, e.g., due to the novelty of the measurement during the first PPT measurement and/or due to a certain degree of familiarization of the following measurements. Therefore, we performed a one-way ANOVA with the pre values of the four measurements of each landmark. Results reveal that no differences occurred meaning that all pre values were not different before the three exercises and the control session. This is described in the results section in lines 216ff.

Comment 5: Considering the upper mentioned critics, I am afraid to say that I suppose this study has lots of methodological limitations so that making it inappropriate to be considered for publication in Biology (ISSN 2079-7737).

Answer 5: Thank you for your valuable work and suggestions. We hope that we clarified and explained the above-mentioned comments. Resulting from the adjustments made, we hope that the scientific quality of our manuscript improved and we are of the opinion that our study should be considered for publication in the respective journal.

References used in this response letter to review III:

[1] Hoffman, Martin D., et al. "Intensity and duration threshold for aerobic exercise-induced analgesia to pressure pain." Archives of physical medicine and rehabilitation 85.7 (2004): 1183-1187.

[2] Rice, David, et al. "Exercise-induced hypoalgesia in pain-free and chronic pain populations: state of the art and future directions." The Journal of Pain 20.11 (2019): 1249-1266.

[3] Koltyn, Kelli F. "Exercise-induced hypoalgesia and intensity of exercise." Sports medicine 32.8 (2002): 477-487.

[4] Vaegter, Henrik Bjarke, et al. "Hypoalgesia after bicycling at lactate threshold is reliable between sessions." European Journal of Applied Physiology 119.1 (2019): 91-102.

[5] Hansen, Simon, et al. "Hypoalgesia after exercises with painful vs. non-painful muscles in healthy subjects–a randomized cross-over study." Scandinavian Journal of Pain 22.3 (2022): 614-621.

[6] Katz-Betzalel, Noa, Irit Weissman-Fogel, and Einat Kodesh. "Aerobic Upper-Limb Exercise-Induced Hypoalgesia: Does It Work?." Applied Sciences 12.22 (2022): 11391.

Round 2

Reviewer 1 Report

It should be recognized that this time the authors took my comments responsibly and made changes to the text of the manuscript, as well as more attentive to the responses to comments. As a result, the design of the study and its possible limitations have become more understandable to readers. I have no other comments. There is only a small remark - after all, the corrections made should be highlighted with a colored marker out of respect for the reviewer and to save his time.

Author Response

Reviewer I

Comment 1: It should be recognized that this time the authors took my comments responsibly and made changes to the text of the manuscript, as well as more attentive to the responses to comments. As a result, the design of the study and its possible limitations have become more understandable to readers. I have no other comments. There is only a small remark - after all, the corrections made should be highlighted with a colored marker out of respect for the reviewer and to save his time.

Answer 1:

Dear reviewer, thank you for you work and effort to review this manuscript.

Reviewer 3 Report

Hello there,

Back to the critics I have already provided for the methodology (the design, and the required number of participants for this design and etc.. ) of this manuscript, the justification provided by the authors does not look sensible for me at all. I therefore would not change my mind about it. I would not recommend to publish this manuscript due to the critics I have already provided.

Best,

Author Response

Reviewer III

Comment 1:

Hello there, Back to the critics I have already provided for the methodology (the design, and the required number of participants for this design and etc.. ) of this manuscript, the justification provided by the authors does not look sensible for me at all. I therefore would not change my mind about it. I would not recommend to publish this manuscript due to the critics I have already provided.

Best,

Answer 1:

Dear reviewer,

thank you for your comment and we will try to discuss the raised concerns by providing more explanations as well as references to original studies for justifying our methodological approach. Apart from this, we would have been happy when some explanations were provided on why our justifications and explanations were considered as “not sensible”. Then, we would have had the opportunity to provide a more nuanced and more specific response. Yet, we are happy to provide some more general remarks regarding this comment as well as the previous comments made:

First, it was noted in the previous comment 1 that the study design was not “balanced” as differently long exercise sessions (resulting in different total work performed) are compared with each other. We would like to stress again that this was exactly the aim of this study, i.e., does a difference in exercise duration (i.e., 30., 45, and 60 minutes) by at the same time remaining the same individual intensity (75% of participants’ VO2max) result in different effects on pain perception? We also stated this aim in lines 7ff of our manuscript. Further, we would like to provide some references of published studies that used the same idea in their study designs: Kodesh and Weissman-Fogel (2014) performed two sessions consisting of (1) an aerobic-continuous exercise (70% heart rate reserve (HHR)) and (2) an interval exercise (4 × 4 min at 85% HRR and 2 min at 60% HRR between cycles), while both exercises lasted for 30 minutes. In this case, the total work performed is obviously also different. Also in this study, this was exactly the aim, which the researchers stated in their paper: “Therefore, the purpose of this study was to compare the analgesic effects of long bouts of high-intensity (4 min) interval exercise at 85% heart rate reserve (HRR) with continuous exercise at 70% HRR” [1]. Another study by Hofmann et al. (2004) conducted three exercise sessions with different intensities and durations. More specifically, participants conducted three treadmill exercises consisting of (1) 10 minutes at 75% VO2max, (2) 30 minutes at 50% VO2max, and (3) 30 minutes at 75% VO2max. Again, the total work performed was different between these three sessions. Again, the authors’ aim was the following: “Therefore, our purpose was to assess pain perception using a pressure-pain stimulus with a pain rating scale before and after exercise of various intensities and durations” [2]. Lastly, a recent study published by the authors in the European Journal of Applied Physiology (2022) also compared four identically long (30 minutes) sessions with each other, while the intensities were different (60, 80, 100, and 100% of the participants’ anaerobic threshold) [3]. Based on these considerations, we believe that the design chosen is appropriate for the research question stated.

Besides, the reviewer noted in the previous comment as well as in this round of revision that the number of participants might have been too low. We would like to provide some explanations for the number of participants chosen. We conducted an a-priori sample size calculation using a medium effect size to estimate the sample as done before [3]. Besides, when looking at the design of this study, every participant conducted 4 experimental sessions and in addition to one pre-experimental test. Therefore, every participant visited our laboratory 4 times due to the cross-over design of the study. This resulted in a total of 144 study sessions, and we are of the opinion that this number is above the average of original studies published in the scientific context of exercise induced hypoalgesia (IEH). We would like to provide some specific examples: Vaegter et al. (2018) conducted a study including 34 participants. In this study, every participant visited the laboratory twice, resulting in 68 study sessions. This study was published in the journal “Pain Medicine” [4]. Kodesh and Weissman-Fogel (2014) published a study in the journal “Applied Physiology, Nutrition, and Metabolism” conducted with 29 participants. In this study, the participants visited the laboratory three times resulting in 87 study sessions [1]. A recent study by Katz-Betzalel et al. (2022) published in the journal “Applied Science” included 31 participants, who visited the laboratory two times, resulting in 62 study sessions [5]. Another study by Vaegter et al. (2017) published in the “European Jounral of Pain” the authors conducted a study including 20 participants. In this study, every participant visited the laboratory three times, resulting in 60 study sessions [6]. These are just some examples of studies in the context of EIH published in highly renowned journals. Based on these considerations, we believe that the number of participants was well chosen.

Second, the reviewer pointed out that the intensity chosen, i.e., 75% of the individuals VO2max, was too low to induce any hypoalgesia. Again, we believe that this intensity is not too low for mainly two reasons: (1) The performance related parameters that were documented (i.e., heart rate, RPE, and lactate) revealed that the intensity was moderate but not generally too low. More specifically, heart rate ranged between 162.5 and 167.8 bpm, RPE between 15.6 and 16.7, and lactate levels between 3.0 and 4.2 mmol/L at the end of the three exercise sessions, respectively (see Table 2 in the manuscript). (2) Other studies also used this intensity in their study designs to evaluate EIH and we would like to refer to our response given in the first round of revision for specific references.

Third, regarding the time span in-between the respective sessions (see previous comment 3), it must be mentioned that this time span is usually used in the field of EIH research. In our previous response in the first revision, we already provided several references to underpin our decision. Yet, we would like to underline that with one more quote of a paper by Vaegter et al. (2018) published in the Journal Pain Medicine: “This time frame [7 days] was chosen to minimize potential carry-over effects from the pain sensitivity assessments and exertion after physical exercise between sessions […]” [4]. Therefore, we are of the opinion that the time span in-between the session was well chosen.

Forth, the reviewer referred to the fact that multiple PPT measurements at on day might have biased the outcome (see previous comment 4). Again, we would like to bring forward that only the control session preceded the 30-minute exercise session. This was the only sequence in which two measurements were conducted at the same day. This decision was made to reduce the number of days, which the participants needed to visit the laboratory and was justified by the assumption that the control session does not induce any hypoalgesia. This was confirmed via the one-way ANOVOA conducted comparing the pre-values of each session to each other (see lines 217ff) as well as via the fact that the control session did not induce any changes in pain perception (see lines 229f and Figure 2). To test whether the pre values of the control session were different to the pre values of the 30-minute session, we now calculated paired t-tests for each landmark regrading PPT values. Results reveal that no significant differences occur between the pre values of the two sessions at any of the eight landmarks supporting the results of the one-way ANOVA presented in the manuscript (see lines 217ff). Therefore, the assumption that multiple pain measurements at one day had a major influence on pain perception can be neglected. Please also refer to the previous response to this comment. However, we are happy to acknowledge this issue in the limitations section and included one new sentence into the manuscript (see lines 378ff).

References used in this response letter

  1. Kodesh, E.; Weissman-Fogel, I. Exercise-induced hypoalgesia - interval versus continuous mode. Appl. Physiol. Nutr. Metab. 2014, 39, 829–834, doi:10.1139/apnm-2013-0481.
  2. Hoffman, M.D.; Shepanski, M.A.; Ruble, S.B.; Valic, Z.; Buckwalter, J.B.; Clifford, P.S. Intensity and duration threshold for aerobic exercise-induced analgesia to pressure pain. Arch. Phys. Med. Rehabil. 2004, 85, 1183–1187, doi:10.1016/j.apmr.2003.09.010.
  3. Tomschi, F.; Lieverkus, D.; Hilberg, T. Exercise-induced hypoalgesia (EIH) in response to different exercise intensities. Eur. J. Appl. Physiol. 2022, doi:10.1007/s00421-022-04997-1.
  4. Vaegter, H.B.; Dørge, D.B.; Schmidt, K.S.; Jensen, A.H.; Graven-Nielsen, T. Test-Retest Reliabilty of Exercise-Induced Hypoalgesia After Aerobic Exercise. Pain Med. 2018, 19, 2212–2222, doi:10.1093/pm/pny009.
  5. Katz-Betzalel, N.; Weissman-Fogel, I.; Kodesh, E. Aerobic Upper-Limb Exercise-Induced Hypoalgesia: Does It Work? Applied Sciences 2022, 12, 11391, doi:10.3390/app122211391.
  6. Vaegter, H.B.; Hoeger Bement, M.; Madsen, A.B.; Fridriksson, J.; Dasa, M.; Graven-Nielsen, T. Exercise increases pressure pain tolerance but not pressure and heat pain thresholds in healthy young men. Eur. J. Pain 2017, 21, 73–81, doi:10.1002/ejp.901.

Round 3

Reviewer 3 Report

Hello there,

Back to the previous critics that I have already provided, and considering the provided revised version along with the cover letter in which they addressed the referees’ comments, I am afraid to mention that I believe this manuscript has not been sufficiently improved to warrant publication in Biology. 

I finally believe this study has lots of methodological limitations so that making it inappropriate to be considered for publication in Biology.

Best regards,